# MIRACL: A ROBUST FRAMEWORK FOR MULTI-LABEL LEARNING ON NOISY MULTIMODAL ELECTRONIC HEALTH RECORDS

## ABSTRACT

Multimodal Electronic Health Records (EHRs), comprising structured time-series data and unstructured clinical notes, offer complementary views of patient health. However, multi-label prediction tasks on multimodal EHR data, such as phenotyping, are hindered by potential label noise, including false positives and negatives. Existing noisy-label learning methods, often designed for single-label vision data, fail to capture real label-dependencies or account for the cross-modal, longitudinal nature of EHRs. To address this, we propose MIRACL (**M**ultimodal **I**nstance **R**elabelling **A**nd **C**orrection for multi-**L**abel noise (MIRACL[1])), a novel framework that systematically addresses these challenges. Notably, MIRACL is the first framework designed to explicitly leverage longitudinal patient context to resolve more challenging multi-label noise scenarios. To achieve this, MIRACL unifies three synergistic mechanisms: (1) a difficulty- and rank-based metric for robust identification of noisy instance-label pairs, (2) a class-aware correction module for robust label refinements, promoting the recovery of real label-dependencies, and (3) a patient-level contrastive regularization loss that leverages both cross-modal and longitudinal patient context to correct for noisy supervision across different visits. Extensive experiments on large-scale multimodal EHR datasets (MIMIC-III/IV) demonstrate that MIRACL achieves state-of-the-art robustness, improving test mAP by over 2% under various noise levels.

## 1 INTRODUCTION

Electronic Health Records (EHRs) data are usually gathered from multimodal sources, providing complementary views of a patient's health. This includes structured data such as temporal medical records (vital signs and lab test results) and unstructured data such as clinical notes (symptom descriptions and the reason for symptoms). Combining both modalities is crucial: while structured data reflect objective physiological signals, unstructured notes capture nuanced physician interpretations, such as symptom reasoning or context, underscoring the need for reliable multimodal fusion. A fundamental task in this domain is multi-label prediction (e.g., Phenotyping), where each patient can exhibit multiple conditions simultaneously. This requires models that can handle semantics of multi-label and heterogeneous modality. Multi-label noise in multimodal EHRs additionally presents a major obstacle to reliable multi-label prediction. Table 1 illustrates these challenges by providing two patient examples. The goal is to learn a robust multi-label model to predict the correct diagnoses (unobserved ground truth) instead of noisy diagnoses (observed but noisy labels).

**Challenge 1: Learning from noisy single-label.** Noisy prediction for patient `P1001` suffers from straightforward Flip Noise, where a Bipolar Disorder label is missing despite clear evidence in the notes (a *false negative*), while a Shock label is added without any supporting evidence from EHR or clinical notes (a *false positive*).

**Challenge 2: Corrupted dependency enforcement under label noise.** Patient `P1002`'s first visit (row 2) demonstrates that the model reinforces a dependency that is corrupted by noisy labels: the flipped $\tilde{y}_{\text{Diabetes}} = 0$ (i.e., the noisy label incorrectly marks Diabetes as absent) distorts the Diabetes $\leftrightarrow$ Hypertension association learned during training. Consequently, despite strong evidence

---

[1] https://github.com/anon-coder-def/MIRACL

| Patient | Key EHR | Note Summary | Ground Truth | | Noisy Labels | | Noisy Prediction | |
|---------|---------|--------------|--------------|---|--------------|---|------------------|---|
| `P1001` | • HR=97
• SpO2=95%
• SBP=95 | • **Bipolar**, **Heroin abuse**.
• No abnormal vitals noted. | 1 | Bipolar | 0 | Bipolar | **0** | Bipolar |
| | | | 0 | Shock | 1 | Shock | **1** | Shock |
| | | | 0 | Sepsis | 0 | Sepsis | **0** | Sepsis |
| `P1002` | • HR=85
• SpO2=96%
• **SBP=160** | • Currently on medication for **diabetes**. | 1 | Diabetes | 0 | Diabetes | **0** | Diabetes |
| | | | 0 | Bipolar | 1 | Bipolar | **1** | Bipolar |
| | | | 1 | Hypertension | 1 | Hypertension | **0** | Hypertension |
| `P1002` | • HR=90
• SBP=125 | • Follow-up for **uncontrolled Diabetes**.
• **Antihypertensive medication**. | 1 | Diabetes | 1 | Diabetes | **1** | Diabetes |
| | | | 0 | Bipolar | 0 | Bipolar | **0** | Bipolar |
| | | | 1 | Hypertension | 0 | Hypertension | **0** | Hypertension |

Table 1: Phenotyping examples of two patients. Noisy Prediction represents the predictions by FlexCare Xu et al. (2024a) using a training set contaminated by Symmetric Flip Noise. Colorbox: Noisy Labels (Observed) against Ground Truth (Unobserved) ( red : false positive cases, blue : false negative cases, green : true cases). Font color: *Noisy Prediction* against *Ground Truth* (**red**: incorrect, **green**: correct). The 1/0 denotes the binary status of a label.

(SBP=160) and a correctly observed (noisy) label for Hypertension, the model under-predicts Hypertension (0). Thus, dependent label noise propagates errors across labels by enforcing corrupted inter-label structure rather than failing to learn any dependency.

**Challenge 3: Synthesizing fragmented evidence across both modalities and patient visits.** The second visit of Patient `P1002` (row 3) presents a complex inference problem. The ground truth, Hypertension is contaminated by a false negative. Correctly inferring this condition requires a model to perform longitudinal cross-modal reasoning: the model must integrate historical numerical evidence from Visit 1's EHR (an SBP of 160, which meets the clinical threshold for hypertension) with current textual evidence from Visit 2's note, which mentions antihypertensive medication.

Together, these examples highlight the necessity for robust multimodal multi-label learning methods that can: (1) correct both positive and negative label errors with high precision, (2) restore the underlying structure of clinical comorbidities from the noisy labels, and (3) leverage complementary information across both modalities and longitudinal patient information.

Existing research in noisy label learning either focuses on single-label image classification Han et al. (2018); Chen et al. (2019), or adopts global reweighting schemes Arazo et al. (2019a). While some recent methods explore multi-label noise Li et al. (2022b); Ghiassi et al. (2023); Xu et al. (2024b) or targeted multimodal medical models Zhang et al. (2022); Hayat et al. (2022); Xu et al. (2024a), they lack a unified mechanism to simultaneously (1) perform efficient instance-level correction to enable the learning of real label-dependencies, and (2) learn a robust model tailored to multimodal EHR data.

To bridge these gaps, we propose **MIRACL**: a Multimodal Instance Relabeling And Correction framework for noisy multimodal multi-label EHR data. We are the first to systematically address multi-label noise in multimodal EHRs by unifying three critical modules: patient-level contrastive loss, class-aware sample selection, and label correction. The main novelties and contributions are:

• We design a class-specific correction module that mitigates the bias toward negative labels and corrects noisy labels to learn correct label dependencies. (Addressing Challenge 1 and Challenge 2).

• We propose a patient-level contrastive regularization loss that promotes generating a cross-modal and longitudinal representation for each patient, alleviating the impact of label noise under high-noise scenarios. (Addressing Challenge 3).

• MIRACL demonstrates its state-of-the-art (SOTA) performance on EHR datasets (MIMIC-III/IV) under different levels and types of multi-label noise.

## 2 RELATED WORK

**Multimodal Multi-Label Learning for Healthcare** Existing multi-label models for healthcare are often embedded in multimodal multitask learning, as in FlexCare Xu et al. (2024a); or are designed for addressing missing modalities, as in M3Care Zhang et al. (2022) by imputing information from similar patients; or originates from multimodal fusion models, as in MedFuse Hayat et al. (2022). However, none of the existing multimodal multi-label healthcare models considers the detrimental effect of label noise.

**Learning from Multi-Label Noise** The traditional approach for handling multi-label is Binary Cross-Entropy (BCE), which treats positive and negative samples with equal weights. To better address imbalance, Focal Loss Lin et al. (2017) assigns different weights to positive and negative samples. ASL Ridnik et al. (2021a) adjusts the weighting scheme asymmetrically by shifting label probabilities, effectively avoiding the contribution of negative labels with extremely low probabilities. MLLSC Ghiassi et al. (2023) is designed for missing and corrupted labels by leveraging loss value for true positive or false positive labels. Other involve estimating transition matrix by leveraging label correlation for clean posterior calculation as in Multi-T Li et al. (2022b). iLaCo Xu et al. (2024b) proposes an instance-level pair correction re-training strategy tailored for noisy multi-label text classification, while failing to scale to large-scale multimodal datasets due to extra re-training. BalanceMix Song et al. (2024) is proposed to handle multi-label noise and imbalance via Mixup-based augmentation; however, it is not directly applicable to multimodal EHR data. Thus, none of the existing noisy multi-label learning methods considers the case on large-scale multimodal EHR datasets. Additional discussion on recent work is provided in Appendix A.7. In contrast to the above methods, we propose an efficient sample-selection-based label correction method in response to all genres of multi-label noise for multimodal data. By leveraging a patient-level contrastive regularization module, we further extend its adaptability to multimodal EHR data.

## 3 METHODOLOGY

### 3.1 OVERVIEW

Overall, the proposed model contains three essential components, as shown in Fig. 1:

- The **Class-Wise Sample Selection Module**: aims to calculate selection criteria $Z$ based on instance dynamics and fit a 2-component Gaussian Mixture Model (GMM) to divide samples into three categories, which are clean sets, uncertain sets, and noisy sets, preparing for correction at the next stage.

- The **Correction Module**: aims to correct the observed noisy label leveraging both label correlation and the probability of being a noisy label from the mixture model.

- The **Patient-Level Contrastive Learning Module**: aims to generate a robust multimodal representation by adding patient-level contrastive regularization loss.

### 3.2 PROBLEM FORMULATION

The multi-label learning task of multimodal data is formally defined as follows. Assuming there is a noiseless dataset $\mathcal{D} = \{(X_i, Y_i, P_i, S_i)\}_{i=1}^N$, where $N$ is the number of instances, $X_i = \{x_i^m\}_{m \in M}$ represents the input data of instance $i$ from modality $m$, from a set of modalities $M$. $Y_i = \{y_i^l\}_{l=1}^L$, where $L$ represents total number of classes; $y_i^l = 1$ represents the presence of class label $l$ for instance $i$ as ground truth; $y_i^l = 0$ otherwise. $S_i$ denotes the unique stay identifier corresponding to instance i. $P_i$ denotes the patient identifier associated with the same instance. In practice, the ground-truth label sets often contain substantial label noise, leading to a noisy dataset defined as $\tilde{\mathcal{D}} = \{(X_i, \tilde{Y}_i, P_i, S_i)\}_{i=1}^N$ where $\tilde{Y}_i = \{\tilde{y}_i^l\}_{l=1}^L$ is the observed noisy label set. Our *objective* is to design a robust model $f^*$ to minimize the empirical risk of the model prediction sets $\hat{Y}_i$ with respect to the latent true label sets $Y_i$, rather than the noisy label sets $\tilde{Y}_i$.

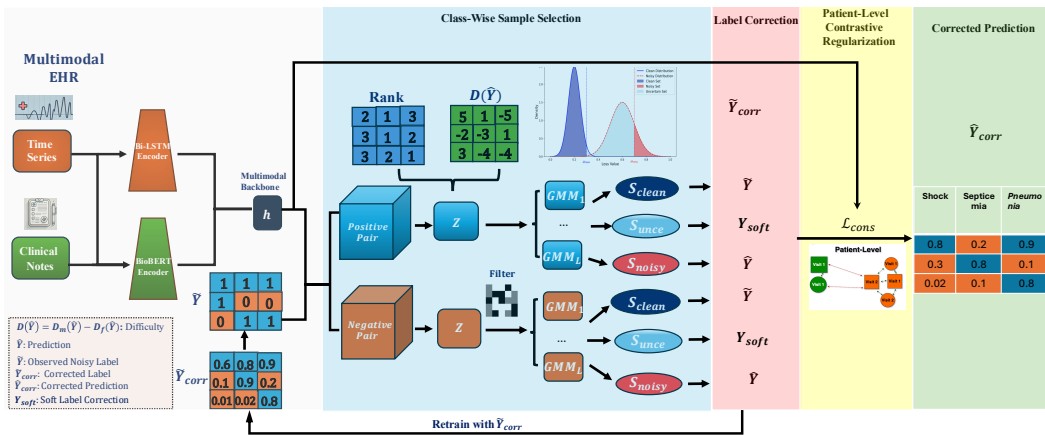

Figure 1: Overall architecture of MIRACL

## 3.3 CLASS-WISE SAMPLE SELECTION STRATEGY

### 3.3.1 MEMORIZATION AND FORGETTING IN NOISY MULTI-LABEL LEARNING

To better distinguish clean and noisy pairs at the instance-pair level without relying on interference of noisy label dependencies, we define the memorization difficulty $D_m(\hat{y}_i^l)$ and forgetting difficulty $D_f(\hat{y}_i^l)$, based on the observation that clean instance-label pairs are typically easier to memorize and harder to forget Hu et al. (2023). To estimate how easy or difficult a label is to learn, we track how often it is memorized and then forgotten during training. For each instance $x_i$ and label $\hat{y}_i^l$ (predicted value for label $l$), we define the overall difficulty $D(\hat{y}_i^l)$ as:

$$D(\hat{y}_i^l) = D_m(\hat{y}_i^l) - D_f(\hat{y}_i^l), \tag{1}$$

where $D_m(\hat{y}_i^l) = \sum_{t=1}^{T} \left| \Delta_m^{(t)}(\hat{y}_i^l) \right|$ as memorization difficulty over the total number of training epochs $T$, $D_f(\hat{y}_i^l) = \sum_{t=1}^{T} \left| \Delta_f^{(t)}(\hat{y}_i^l) \right|$ as forgetting difficulty, $\Delta_m^{(t)}(\hat{y}_i^l)$ is an indicator function that equals 1 if label $\hat{y}_i^l$ was *incorrectly predicted at epoch $t-1$* but *correctly predicted at epoch $t$* — representing a memorization event. Conversely, $\Delta_f^{(t)}(\hat{y}_i^l)$ equals 1 if $\hat{y}_i^l$ was *correctly predicted at epoch $t-1$* but *incorrectly predicted at epoch $t$* — representing a forgetting event. Clean instance-label pairs typically exhibit lower $D_m(\hat{y}_i^l)$, indicating they are memorized quickly and stably, whereas noisy pairs tend to have higher memorization difficulty. By focusing on the transition dynamics of individual label predictions, the overall difficulties provide a label-wise estimation of noise.

**Selection Metric** Considering the inter-dependencies inherent in multi-label learning, the rank of an instance has been shown to effectively capture inter-label relationships without being significantly disrupted by noisy instance-label pairs Xu et al. (2024b). Motivated by this, we introduce a multi-label selection metric $Z(\tilde{y}_i^l)$ for the $i$-th instance and $l$-th label, which satisfies two key properties: 1) Leveraging reliability from a single-label perspective; 2) Identifying noisy signals by capturing inter-label dependencies through instance-level prediction dynamics. The selection metric is defined as,

$$Z(\tilde{y}_i^l) = \alpha \operatorname{Rank}(\hat{y}_i^l) + (1 - \alpha)(D_m(\tilde{y}_i^l) - D_f(\tilde{y}_i^l)), \tag{2}$$

where $\alpha$ balances the reliance on ranking-based selection versus memorization-forgetting difficulty. We set $\alpha = 0.5$ to balance the contribution of both signals. $\operatorname{Rank}(\hat{y}_i^l)$ is the rank of label confidence from model predictions using the rank function $\operatorname{Rank}(.)$, which is highly indicative of clean positive labels. A higher $Z(\tilde{y}_i^l)$ score suggests that label $\tilde{y}_i^l$ is more likely to be corrupted, enabling the model to dynamically filter out noisy labels.

**Sample Selection** To allow the model to apply distinct correction strategies based on the estimated label reliability, we use three-way partitioning to improve robustness under varying noise levels. For this task, we model the normalized selection metric $\tilde{Z}$ per-class and per-label using GMM,

$2 \times L$ in total. This allows us to statistically separate samples, with each component modeling clean or noisy samples for that category. First, we normalize the distribution $Z$ (Eq. 2) respectively, using $\tilde{Z}^{lc} = \frac{Z^{lc} - \min(Z^{lc})}{\max(Z^{lc}) - \min(Z^{lc}) + \epsilon}$, to ensure the distribution values fall in the range from 0 to 1. $l$ denotes the index of label; $c \in \{0, 1\}$ indicates the binary value of a class label; $\epsilon$ is set as $1e^{-6}$ to prevent zero division errors. We partition samples into clean, uncertain, and noisy sets using the mean of each component based on empirical observation and Lu & He (2022). To avoid hand-crafted thresholds that may not generalize across noise levels, we then adopt a thresholding strategy inspired by Huang et al. (2022).

Specifically, for each label $l$ and the value of the class label $c$, we model the normalized selection score $\tilde{Z}^{lc}$ using a bimodal GMM. Let $\mu_{\text{clean}}^{lc}$ and $\mu_{\text{noisy}}^{lc}$ denote the mean values of the two mixture components, with $\mu_{\text{clean}}^{lc} < \mu_{\text{noisy}}^{lc}$. Based on these thresholds, we further classify normalized selection score set $\mathcal{S} = \{\tilde{Z}^{lc}\}$ into three subsets:

$$
\begin{aligned}
\mathcal{S}_{\text{clean}} &= \{\tilde{Z}^{lc} \mid \tilde{Z}^{lc} \leq \mu_{\text{clean}}^{lc}\}, \\
\mathcal{S}_{\text{unce}} &= \{\tilde{Z}^{lc} \mid \mu_{\text{clean}}^{lc} < \tilde{Z}^{lc} < \mu_{\text{noisy}}^{lc}\}, \\
\mathcal{S}_{\text{noisy}} &= \{\tilde{Z}^{lc} \mid \tilde{Z}^{lc} \geq \mu_{\text{noisy}}^{lc}\}.
\end{aligned}
\tag{3}
$$

Pairs with selection score falling below $\mu_{\text{clean}}^{lc}$ (in $\mathcal{S}_{\text{clean}}$) are treated as clean and used directly for training, while those above $\mu_{\text{noisy}}^{lc}$ (in $\mathcal{S}_{\text{noisy}}$), as well as uncertain samples in between (in $\mathcal{S}_{\text{unce}}$), are handled by tailored noise mitigation strategies.

To reduce computational overhead and ensure reliable fitting across all $2 \times L$ GMMs, we **fit GMMs per epoch** [2], which both accelerates training and provides more diverse samples for stable convergence.

### 3.4 JOINT LABEL CORRECTION

Two types of noise occur in noisy multi-label learning: false positive noise and false negative noise. To address these issues, we have designed dedicated correction strategies for each. We refer to a set of instance-label pairs with negative/positive labels as *negative/positive pairs*.

**Clean Set**: Pairs with scores in $\mathcal{S}_{\text{clean}}$, which are likely to be clean, the model should improve its trustworthiness by using the original label without performing any label correction.

**Uncertain Set**: Pairs with scores in $\mathcal{S}_{\text{unce}}$, we apply soft label correction by interpolating between the model's prediction and the original label, inspired by Arazo et al. (2019a). The interpolation weight is derived from the uncertainty score $U^{lc}$ based on the class-wise GMM. Specifically,

$$
U^{lc} = \frac{\tilde{Z}^{lc} - \mu_{\text{clean}}^{lc}}{\mu_{\text{noisy}}^{lc} - \mu_{\text{clean}}^{lc} + \epsilon},
\tag{4}
$$

where $U^{lc}$ is clipped in the range $[0, 1]$. We then define soft label correction as:

$$
Y_{\text{soft}}^{lc} = U^{lc} \cdot \hat{Y}^{lc} + (1 - U^{lc}) \cdot \tilde{Y}^{lc}.
\tag{5}
$$

which is the expectation of the ground truth label for a particular uncertain sample. A lower uncertainty score means the label is highly likely to be the original annotation, and vice versa. This strategy is particularly effective in handling samples within the ambiguous decision boundary, allowing the model to dynamically adjust the impact of clean and predicted labels during training.

**Noisy Set**: For pairs with scores in $\mathcal{S}_{\text{noisy}}$, the model should trust the prediction and perform soft label correction.

**Negative Pairs**: Real-world datasets often contain many true negative pairs, which can significantly distract the model from accurately identifying false negative cases, as illustrated in Xu et al. (2024b). In addition to label correction, we apply a filtering mechanism to retain correlated negative labels against positive labels. This filtered dataset $\mathcal{Z}^-$ helps retain informative negative pairs while also improving computational efficiency. Specifically, we define the filtered score set $\mathcal{Z}^-$ as:

$$
\mathcal{Z}^- = \{\tilde{Z}^{lc} | \mathbf{S} > \tau\}, \mathbf{S} = \tilde{Y}\mathbf{C}
\tag{6}
$$

---

[2]Computational Analysis is provided in Appendix A.6

where $\mathbf{S}$ reflects how strongly each label is supported by correlated labels in the prediction space, $\tau$ is the correlation coefficient threshold, $\mathbf{C}$ is the correlation matrix between the observed labels that captures how often label $k$ co-occurs with label $j$ (Appendix D). The corrected label for negative pairs, $\tilde{Y}_{\text{corr}}^{l0}$ is defined as follows:

$$\tilde{Y}_{\text{corr}}^{l0} = \begin{cases} \tilde{Y}^{l0}, & \tilde{Z}^{l0} \in \mathcal{S}_{\text{clean}} \cap \mathcal{Z}^- \\ Y_{\text{soft}}^{l0}, & \tilde{Z}^{l0} \in \mathcal{S}_{\text{unce}} \cap \mathcal{Z}^- \\ \hat{Y}^{l0}, & \tilde{Z}^{l0} \in \mathcal{S}_{\text{noisy}} \cap \mathcal{Z}^- \end{cases}, \tag{7}$$

where $\hat{Y}^{lc}$ represents the model prediction for label $l$ and the observed class $c$.

**Positive Pairs**: Due to the shortage of positive pairs, we propose to consider all positive pairs and correct positive pairs based on the criteria if the selection metric $\tilde{Z}^{lc}$ is less than the mean of the smaller mixture of that particular class $\mu_{\text{clean}}^{lc}$. The corrected labels for positive pairs $\tilde{Y}_{\text{corr}}^{l1}$ are:

$$\tilde{Y}_{\text{corr}}^{l1} = \begin{cases} \tilde{Y}^{l1}, & \tilde{Z}^{l1} \in \mathcal{S}_{\text{clean}} \\ Y_{\text{soft}}^{l1}, & \tilde{Z}^{l1} \in \mathcal{S}_{\text{unce}} \\ \hat{Y}^{l1}, & \tilde{Z}^{l1} \in \mathcal{S}_{\text{noisy}} \end{cases}. \tag{8}$$

### 3.5 CROSS-MODAL CONTRASTIVE REGULARIZATION

Contrastive learning has proven effective for learning robust multimodal representations by pulling together positive pairs and pushing apart negative ones Li et al. (2022a). To avoid confusion, we note that the term *positive/negative pair* here refers to cross-modal representations of the same instance. A key challenge in multimodal EHRs lies in defining what constitutes a positive pair. A common strategy— Visit-Level contrastive learning — treats different modalities (e.g., structured EHR and clinical notes) from the same hospital visit as positives, and data from all other visits, even from the same patient, as negatives. As shown on the left side of Fig. 2, this enforces align-

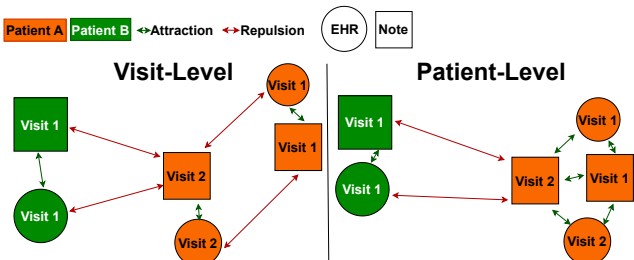

Figure 2: Conceptual illustration of Patient-Level vs. Visit-Level contrastive learning. While a Visit-Level approach (left) only aligns modalities within a single visit, our Patient-Level strategy (right) correctly leverages patient identity to group all representations from the same patient (orange), while separating them from a different patient (green).

ment only within a single visit, ignoring the longitudinal nature of patient records. It fails to capture stable patient identity across admissions and cannot leverage cross-visit evidence to resolve noise.

#### 3.5.1 PATIENT-LEVEL MULTIMODAL CONTRASTIVE REGULARIZATION

To address this, we propose a Patient-Level Multimodal Contrastive Loss, which softly aligns all representations from the same patients — across visits and modalities — into a shared embedding space. Within a mini-batch, we define positive and negative pairs based on *patient identity*. For an anchor (e.g., an EHR embedding $\mathbf{h}_i$ from patient $P_A$), its positive set $P(i)$ includes: 1) its cross-modal counterpart from the same visit (e.g., a note embedding), and 2) all other representations from $P_A$, regardless of modality or visit. All representations from any other patients $P_B$ form the negative pairs.

**Contrastive Loss:** We apply this contrastive loss on latent embeddings extracted from modality-specific encoders: a bidirectional LSTM Graves et al. (2005) to model sparse and temporal dependencies in multivariate EHR sequences, and a pretrained BioBERT Lee et al. (2019) for capturing domain-specific semantics from clinical notes. This allows our model to learn cross-modal and cross-visit consistency even under noisy supervision.

We adapt the SCL Li et al. (2022a) to encourage patient-level alignment across modalities and visits. Specifically, we concatenate the latent representations from both modality-specific encoders—structured EHR ($\mathbf{h}^{(E)}$) and clinical notes ($\mathbf{h}^{(N)}$)—into a unified set: $\mathbf{H} = [\mathbf{h}_1^{(E)}, \ldots, \mathbf{h}_B^{(E)}, \mathbf{h}_1^{(N)}, \ldots, \mathbf{h}_B^{(N)}]$ along with their corresponding patient identifiers, where $B$ is the batch size. The contrastive loss $\mathcal{L}_{\text{cons}}$ for an anchor $\mathbf{h}_i$ is then defined as:

$$\mathcal{L}_{\text{cons}} = -\frac{1}{|P(i)|} \sum_{p \in P(i)} \log \frac{\exp(\text{sim}(\mathbf{h}_i, \mathbf{h}_p)/\tau_{\text{temp}})}{\sum_{k \neq i} \exp(\text{sim}(\mathbf{h}_i, \mathbf{h}_k)/\tau_{\text{temp}})}, \tag{9}$$

where $P(i)$ denotes the set of all other representations in the batch that share the same patient ID as $\mathbf{h}_i$; $\text{sim}(\cdot, \cdot)$ denotes cosine similarity; and $\tau_{\text{temp}}$ is a temperature scaling factor. This loss encourages all embeddings derived from the same patients — across modalities and visits — to cluster in latent space, forming a stable, identity-preserving representation that supports downstream noise correction.

The overall training pipeline of MIRACL, consisting of a warm-up phase and a correction phase, is detailed in Appendix A.4.

Table 2: Comparison of performance on MIMIC-IV Phenotyping test dataset under different noise conditions ($\rho_+$, $\rho_-$). The evaluation metric is average mAP with standard deviation (in bracket) in the last epoch across 3 runs. The best average results are highlighted in **bold**. *, **, and *** indicate $p < 0.05$, $p < 0.01$, and $p < 0.001$.

| Model | Symmetric Flip Noise (%) | | Asymmetric Flip Noise (%) | | | | Balanced Noise (%) | |
|---|---|---|---|---|---|---|---|---|
| | (20,20) | (40,40) | (0,20) | (0,40) | (20,0) | (40,0) | (20,4.48) | (40,8.96) |
| ASL | 0.501(0.001) | 0.377(0.026) | 0.511(0.004) | 0.471(0.003) | 0.472(0.003) | 0.448(0.006) | 0.474(0.002) | 0.458(0.011) |
| Focal | 0.184(0.000) | 0.182(0.000) | 0.184(0.002) | 0.184(0.001) | 0.188(0.003) | 0.189(0.001) | 0.185(0.002) | 0.185(0.001) |
| GCE | 0.523(0.002) | 0.411(0.007) | 0.538(0.002) | 0.497(0.005) | 0.514(0.003) | 0.446(0.005) | 0.514(0.006) | 0.469(0.002) |
| MLLSC | 0.196(0.011) | 0.202(0.009) | 0.202(0.005) | 0.202(0.005) | 0.207(0.012) | 0.200(0.006) | 0.197(0.006) | 0.199(0.011) |
| MultiT | 0.522(0.000) | 0.391(0.008) | 0.546(0.004) | 0.487(0.007) | 0.555(0.008) | 0.539(0.009) | 0.546(0.003) | 0.517(0.003) |
| MedFuse | 0.433(0.014) | 0.296(0.001) | 0.461(0.009) | 0.418(0.009) | 0.484(0.001) | 0.477(0.002) | 0.471(0.007) | 0.399(0.013) |
| M3Care | 0.432(0.001) | 0.368(0.001) | 0.444(0.001) | 0.434(0.001) | 0.454(0.000) | 0.449(0.001) | 0.449(0.001) | 0.436(0.001) |
| FlexCare | 0.510(0.004) | 0.369(0.020) | 0.542(0.003) | 0.513(0.008) | 0.554(0.005) | 0.542(0.006) | 0.548(0.001) | 0.510(0.004) |
| MIRACL | 0.540(0.002)*** | 0.439(0.012)** | 0.560(0.004)** | 0.539(0.006)* | 0.569(0.003) | 0.557(0.001)* | 0.564(0.002)*** | 0.537(0.001)** |

## 4 EXPERIMENTS

### 4.1 EXPERIMENTAL SETUP

**Datasets**: In this study, we utilize three datasets derived from MIMIC-III Johnson et al. (2016) and MIMIC-IV Johnson et al. (2023a;b), which include two multi-label learning sub-task datasets Phenotyping for MIMIC-III PHE and MIMIC-IV PHE, and MIMIC-IV DIA (Appendix C). We split each dataset into training and test sets using an 8:2 ratio **on a patient-wise basis**. We introduce artificial noise to the training dataset only, while keeping the test set unmodified.

**Noisy Label Generation**: Following prior work Xu et al. (2024b), we simulate three types of label noise to assess model robustness under controlled settings, governed by a noise ratio $\rho$. They are 1) *Symmetric Noise:* Each label is flipped with a uniform probability $\rho$ (i.e., $\rho_+ = \rho_- = \rho$), where $\rho_+$ represents the probability of flipping a label from 1 to 0, and vice versa for $\rho_-$; 2) *Asymmetric Noise:* Labels are flipped with different probabilities for positive and negative pairs ($\rho_+ \neq \rho_-$); 3) *Balanced Flip Noise:* As defined in Xu et al. (2024b), this method ensures a similar number of flips for both positive and negative instances: $\rho_+ = \rho$, $\rho_- = \frac{L_{avg}}{L-L_{avg}}\rho_+$, $L_{avg}$ represents the average number of positive instance-label pairs per dataset.

**Baselines**: We compare our approach to several baseline models for noisy multi-label learning from two categories: 1) *Multimodal multi-label healthcare models:* **M3Care** Zhang et al. (2022), **Med-Fuse** Hayat et al. (2022), **FlexCare** Xu et al. (2024a); 2) *Noisy multi-label methods* (built upon the FlexCare Xu et al. (2024a) backbone): **Focal Loss (Focal)** Lin et al. (2017), **Asymmetric Focal Loss (ASL)** Ridnik et al. (2021a), **Generalized Cross-Entropy (GCE)** Zhang & Sabuncu (2018), **MLLSC** Ghiassi et al. (2023), **MultiT** Li et al. (2022b).

**Evaluation Metric**: Consistent with the previous literature Xu et al. (2024b), we train the model on the *noisy* training set and report the *mean and standard deviation* of mean average precision

(mAP) over three independent runs on the *clean* test set. We evaluate model robustness based on the average performance at the *final training epoch*. We have applied one-sided Student's $t$-tests against the second best baseline (Appendix C).

**Hyperparameter Configuration**: Appendix A.5.

Table 3: Comparison of performance on MIMIC-III Phenotyping test dataset under different noise conditions ($\rho_+$, $\rho_-$). The evaluation metric is average mAP with standard deviation (in bracket) in the last epoch across 3 runs.

| Model | Symmetric Flip Noise (%) | | Asymmetric Flip Noise (%) | | | | Balanced Noise (%) | |
|---|---|---|---|---|---|---|---|---|
| | (20,20) | (40,40) | (0,20) | (0,40) | (20,0) | (40,0) | (20,3.95) | (40,7.90) |
| ASL | 0.462(0.002) | 0.316(0.012) | 0.487(0.001) | 0.450(0.001) | 0.408(0.005) | 0.321(0.004) | 0.431(0.004) | 0.411(0.004) |
| Focal | 0.165(0.000) | 0.166(0.001) | 0.166(0.001) | 0.166(0.001) | 0.166(0.001) | 0.166(0.001) | 0.166(0.000) | 0.166(0.001) |
| GCE | 0.457(0.006) | 0.306(0.031) | 0.481(0.003) | 0.434(0.007) | 0.444(0.005) | 0.326(0.032) | 0.454(0.004) | 0.408(0.004) |
| MLLSC | 0.162(0.010) | 0.155(0.002) | 0.157(0.007) | 0.157(0.001) | 0.158(0.005) | 0.161(0.011) | 0.158(0.008) | 0.158(0.008) |
| MultiT | 0.452(0.007) | 0.309(0.006) | 0.482(0.001) | 0.423(0.002) | 0.493(0.003) | 0.474(0.005) | 0.481(0.005) | 0.445(0.008) |
| MedFuse | 0.331(0.003) | 0.252(0.001) | 0.376(0.005) | 0.326(0.001) | 0.421(0.004) | 0.400(0.015) | 0.390(0.009) | 0.349(0.012) |
| M3Care | 0.382(0.001) | **0.316(0.002)** | 0.392(0.001) | 0.379(0.000) | 0.405(0.003) | 0.401(0.002) | 0.401(0.001) | 0.384(0.001) |
| FlexCare | 0.441(0.006) | 0.298(0.016) | 0.476(0.000) | 0.436(0.008) | 0.491(0.007) | 0.476(0.005) | 0.477(0.004) | 0.436(0.014) |
| MIRACL | **0.469(0.006)**\*\* | 0.279(0.009) | **0.498(0.004)**\*\*\* | **0.471(0.006)**\*\*\* | **0.511(0.002)**\*\*\* | **0.497(0.001)**\*\*\* | **0.504(0.002)**\*\*\* | **0.475(0.001)**\*\*\* |

## 4.2 COMPARATIVE EVALUATION

**MIMIC-IV PHE:** As shown in Table 2, MIRACL achieves statistically significant, strong, and consistent mAP performance across various noise conditions, outperforming other methods by over 2%. This consistent performance demonstrates MIRACL's robustness under all noise types. Notably, MIRACL achieves the largest relative improvements under symmetric noise, outperforming the best baseline by over 3.6% at 40% corruption. Under asymmetric and balanced noise, MIRACL remains the top performer, with stable margins of 1.5%, reflecting its reliability across label corruption.

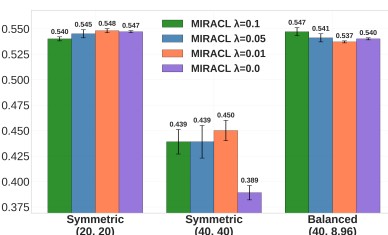

Figure 3: Impact of $\lambda_{\text{cons}}$ on MIMIC-IV.

**MIMIC-III PHE:** As shown in Table 3, MIRACL demonstrates consistently strong performance in a wide range of noise configurations on MIMIC-III. While MIRACL experiences a substantial drop in performance under the higher noise level of Sym. 40%, this is expected given the fixed hyperparameter setup used across datasets. Fig. 3 further illustrates that MIRACL becomes more sensitive to the contrastive regularization strength $\lambda_{\text{cons}}$ under extreme noise. Despite this, MIRACL continues to outperform all baselines in these challenging settings and achieves competitive results even without tuning $\lambda_{\text{cons}}$, highlighting its robustness and practical generalizability.

## 4.3 ABLATION STUDIES

In this section, we empirically evaluate the contributions of different components of our model by ablation study. We select Symmetric $(20, 20)$, Asymmetric $(0, 20)$, and Balanced $(20, 4.48)$ noise as representative cases to assess the effectiveness of different model components. As shown in Table 4, Patient-Level Contrastive Regularization (+ Con. Reg.) yields the largest performance boost, particularly under higher noise ratios, confirming its central role in learning robust representations. Label correction further improves performance, with Rank-based filtering outperforming Difficulty-based selection. The full MIRACL model consistently achieves the best performance, demonstrating the complementary strengths of contrastive regularization and class-aware noise correction.

Table 4: Ablation study of MIRACL under MIMIC-IV.

| MIRACL Variant | (0,20) | (20,20) | (20,4.48) |
|---|---|---|---|
| Baseline | 0.542 ± 0.003 | 0.510 ± 0.004 | 0.548 ± 0.001 |
| + Con. Reg. | 0.553 ± 0.001 | 0.534 ± 0.004 | 0.557 ± 0.001 |
| + Correction | / | / | / |
| w/ Loss only | 0.550 ± 0.002 | 0.530 ± 0.004 | 0.563 ± 0.002 |
| w/ Mems only | 0.557 ± 0.004 | 0.535 ± 0.004 | 0.561 ± 0.002 |
| w/ Rank only | 0.559 ± 0.004 | 0.539 ± 0.006 | 0.561 ± 0.001 |
| MIRACL | **0.560 ± 0.004** | **0.540 ± 0.002** | **0.564 ± 0.002** |

### 4.3.1 PATIENT-LEVEL CONTRASTIVE LEARNING ANALYSIS

We observe that the performance gain from contrastive regularization (Con. Reg.) is most pronounced under high-noise conditions, as shown in Fig. 3. For instance, under Sym. 40% noise, increasing $\lambda_{cons}$ from 0.0 to 0.05 substantially improves mAP, suggesting that contrastive signals become increasingly valuable as label supervision degrades. This is because, when ground-truth labels are unreliable, our patient-level contrastive loss provides an alternative training signal by leveraging structural consistency across modalities and visits. Notably, the performance curve shows a clear upward trend as $\lambda_{cons}$ increases from 0.0 to 0.05 under high noise level, after which it plateaus or slightly declines. In contrast, in low-noise settings (e.g., Sym. 20%), the effect of $\lambda_{cons}$ is relatively mild, reflecting that corrected labels already offer strong supervision. These observations highlight the role of contrastive regularization as an effective fallback mechanism under severe label noise.

### 4.3.2 CORRECTION LABELS ANALYSIS

Fig. 4 illustrates the effectiveness of our label correction strategy by tracking test accuracy trends over 30 training epochs under Sym. 20% noise on MIMIC-IV phenotyping. The blue line indicates accuracy on clean labels, orange indicates accuracy on noisy labels after correction by MIRACL, and green indicates accuracy on noisy labels without any correction. Notably, the accuracy of corrected noisy (orange) consistently and substantially outperforms its uncorrected counterpart (green) throughout training, highlighting the critical role of our correction pipeline in denoising supervision. This performance gap demonstrates that MIRACL's GMM-based filtering and class-aware correction effectively recover useful signal from noisy instance-label pairs. Furthermore, the accuracy on clean data (blue) steadily improves as noisy supervision improves, indicating that our correction not only rescues noisy labels, but also stabilizes overall learning by preventing error propagation from corrupted labels.

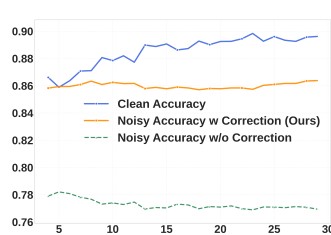

Figure 4: MIMIC-IV (Sym. 20%).

### 4.3.3 SELECTION METRIC ANALYSIS

We analyze the distributions of the selection metrics for MIMIC-IV (Fig. 5). An effective metric should clearly separate clean from noisy pairs. In Table 4, we find that while Overall Difficulty (Fig. 5c) excels at isolating clean pairs at the individual instance-label level based on their learning dynamics. Conversely, Rank (Fig. 5b) implicitly captures inter-label dependencies, as the model learns to assign correlated ranks to clinically related conditions. Although the BCE loss (Fig. 5a) aligns with the small-loss criterion (pairs with small loss tend to be clean Song et al. (2019)), it exhibit similarly low losses in later epochs, making it increasingly difficult to distinguish them from clean ones as training progresses (Appendix B). Therefore, we propose a Holistic Metric ($Z$) that

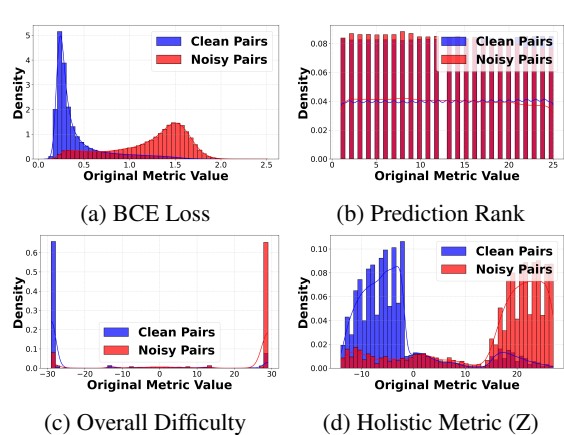

(a) BCE Loss     (b) Prediction Rank

(c) Overall Difficulty     (d) Holistic Metric (Z)

Figure 5: Metric Distribution (Sym. 20% Noise, MIMIC-IV)

fuses these complementary signals. As shown in Fig. 5d, this fusion of pair-level dynamics and instance-level context yields a distinctly bimodal distribution with significantly improved separation between clean and noisy pairs, providing a robust foundation for our GMM-based selection.

## 5 CONCLUSION

In conclusion, we present **MIRACL**, a unified framework that robustly addresses multi-label noise in multimodal EHRs. We introduce a patient-level contrastive regularization loss to support cross-

modal and cross-visit learning, along with a novel selection metric that integrates the strengths of instance-level and rank-based learning to more effectively distinguish clean from noisy instance-label pairs. By fitting class-wise GMMs and jointly training with corrected soft labels, MIRACL achieves state-of-the-art performance on the MIMIC datasets. In future work, we plan to explore cross-modal attention mechanisms to further improve label reliability and extend MIRACL to datasets with additional modalities.

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

## CONTENTS OF APPENDIX

**Reproducibility Statement.**   We have taken multiple steps to ensure the reproducibility of this work. All datasets used (MIMIC-III/IV) are publicly available under credentialed access, and we provide a complete description of preprocessing and cohort construction in Appendix A.2. Details of our proposed model, training protocol, and hyperparameters are included in Appendix A.4 and Appendix A.5. To facilitate replication of our experiments, we will release the full source code and configuration files in the supplementary materials through an anonymous GitHub repository `https://github.com/anon-coder-def/MIRACL`. Upon acceptance, the complete code will be open-sourced and made publicly available on GitHub. We also report results across multiple random seeds and provide ablation studies in Appendix A.5 and Sec 4.3, ensuring that the conclusions are robust to implementation choices.

**LLM Usage Statement.**   We used a general-purpose large language model (OpenAI ChatGPT; accessed Aug–Sep 2025) strictly as a writing assistant. Concretely, it suggested grammar and style edits, helped rephrase sentences for clarity and brevity, and assisted with minor LaTeX formatting (e.g., equation alignment, table spacing). The LLM did *not* contribute to research ideation, problem formulation, method design, theoretical results, experimental setup, code implementation, data analysis, or the writing of technical content such as proofs and related work. All scientific claims, algorithms, and results were authored and verified by the authors, who take full responsibility for the content. No non-public code, proprietary data, or patient-level information were provided to the LLM. The LLM is not listed as an author. This disclosure complies with the conference policy on LLM usage.

# A  EXPERIMENTAL SETUP & IMPLEMENTATION DETAILS

## A.1  PREDICTION TASKS

- Phenotyping(PHE): A multi-label classification problem that classifies which of 25 acute care conditions are present in a given patient ICU stay record.
- Diagnosis(DIA): A multi-label classification problem that predicts 14 diagnosis conditions.

## A.2  DATASET AND PREPROCESSING

### A.2.1  DATASET DOWNLOAD

Please check our GitHub repository `https://github.com/anon-coder-def/MIRACL` for more details. You will first need to request access to download MIMIC dataset.

### A.2.2  PREPROCESSING

Due to the limited number of multimodal multi-label learning datasets, we choose 3 EHR-based datasets to further validate our approach.

- MIMIC-IV (PHE, DIA) Johnson et al. (2023b;a) Phenotyping, Diagnosis: We use the same pre-processing procedure as Xu et al. (2024a) for MIMIC-IV.
- MIMIC-III (PHE) Johnson et al. (2016): We use the same preprocessing procedure as Harutyunyan et al. (2019) for EHR. For clinical notes, we adapt the Khadanga et al. (2019) to extract clinical notes and use a maximum length of 512 for each.

Table 5: Statistics of the Multimodal Multi-Label dataset, $TS$ refers to time series, $T$ refers to Clinical Notes.

| Task | Prediction Task | Modality | # Records | $L$ | $L_{avg}$ |
|---|---|---|---|---|---|
| **MIMIC-IV PHE** | Clinical Phenotype | $\{TS, T\}$ | 59,798 | 25 | 4.575 |
| **MIMIC-IV DIA** | Clinical Diagnosis | $\{TS, T\}$ | 132,576 | 14 | 2.246 |
| **MIMIC-III PHE** | Clinical Phenotype | $\{TS, T\}$ | 41,904 | 25 | 4.126 |

## A.3  BASELINE DESCRIPTION

**Baselines**: We use FlexCare Xu et al. (2024a) as the backbone model for noisy multi-label approaches and compare our approach to several baseline models with same hyperparameter setting:

- **Focal Loss (Focal)** Lin et al. (2017): Addresses class imbalance by focusing more on hard-to-classify examples.
- **Asymmetric Focal Loss (ASL)** Ridnik et al. (2021b): Modifies Focal Loss to better handle label imbalance in multi-label settings by assigning different weights to relevant and irrelevant labels.
- **Generalised Cross-Entropy (GCE)** Zhang & Sabuncu (2018): A robust loss function designed for noisy multi-label classification, combining properties of mean absolute error (MAE) and cross-entropy (CE) for better noise tolerance.
- **MLLSC** Ghiassi et al. (2023): Handles missing and corrupted labels by leveraging loss values for both true-positive and false-positive labels to improve model robustness.
- **MultiT** Li et al. (2022b): Utilises label correlations to estimate a transition matrix for noisy multi-label learning, effectively aligning observed labels with true labels to mitigate label noise.

We also compare our approach against existing multimodal healthcare model:

- **M3Care** Zhang et al. (2022): Proposes an end-to-end multimodal framework that addresses missing modalities in healthcare data by imputing missing modalities information from similar patients.

- **MedFuse** Hayat et al. (2022): A lightweight and flexible multimodal model that projects each modality (e.g., time-series EHR, medical images) into a shared latent space using modality-specific encoders, by a LSTM fusion-based module
- **FlexCare** Xu et al. (2024a): A flexible multimodal multitask framework that decomposes parallel task prediction into asynchronous single-task outputs, uses task-agnostic representation learning with covariance regularization across modalities, and integrates these via a task-guided hierarchical fusion module to support multimodal multi-label EHR prediction.

## A.4 OVERALL TRAINING PROCEDURE

### A.4.1 WARM-UP PHASE:

As demonstrated by Arazo et al. (2019b), cross-entropy loss distribution naturally fits a mixture model with theoretical justification. Similarly, in the multi-label setting, the binary cross-entropy (BCE) loss $\mathcal{L}_{\text{bce}} = -\left(\tilde{Y} \cdot \log(\hat{Y}) + (1 - \tilde{Y}) \cdot \log(1 - \hat{Y})\right)$. We treat $\mathcal{L}_{\text{cons}}$ as a regularization term and incorporate it into the final objective:

$$\mathcal{L}_{\text{warmup}} = \mathcal{L}_{\text{bce}} + \lambda_{\text{cons}} \cdot \mathcal{L}_{\text{cons}}, \tag{10}$$

where $\lambda_{\text{cons}}$ is a weighting coefficient controlling the strength of the contrastive regularization.

### A.4.2 CORRECTION PHASE:

We fit the computed selection scores to Gaussian Mixture Models from the last epoch over the entire dataset. We then perform Label correction based on the sample selection mechanism derived from the GMMs. We train using the corrected BCE loss $\mathcal{L}_{\text{corr}} = \mathcal{L}_{\text{bce}}(\tilde{Y}_{\text{corr}}, \hat{Y})$ for the remainder of the training period: At the beginning of training, the model relies more on the corrected loss to mitigate the influence of label noise. As training progresses and the model learns more robust representations, the weights gradually shifts towards the standard BCE loss, balancing the contributions of both components dynamically.

Mathematically, the weighted loss $\mathcal{L}_{\text{weighted}}$ is defined as:

$$\mathcal{L}_{\text{weighted}} = \beta_t \, \mathcal{L}_{\text{corr}} + (1 - \beta_t) \, \mathcal{L}_{\text{bce}} + \lambda_{\text{cons}} \cdot \mathcal{L}_{\text{cons}}, \tag{11}$$

where $T$ represents max epoch, $\beta_t$ increases linearly with epoch $t$ that transitions smoothly from an initial value $\beta_0 = 1$ to a final value $\beta_f = 0.5$ for stabilising the final stages of training in label correction framework Arazo et al. (2019b). The detailed algorithm is shown below in Algorithm 1.

$$Y_{\text{soft}}^{lc} = U^{lc} \cdot \hat{Y}^{lc} + (1 - U^{lc}) \cdot \tilde{Y}^{lc}. \tag{12}$$

$$\tilde{Y}_{\text{corr}}^{l0} = \begin{cases} \tilde{Y}^{l0}, & \tilde{Z}^{l0} \in \mathcal{S}_{\text{clean}} \cap \mathcal{Z}^- \\ Y_{\text{soft}}^{l0}, & \tilde{Z}^{l0} \in \mathcal{S}_{\text{unce}} \cap \mathcal{Z}^- \\ \hat{Y}^{l0}, & \tilde{Z}^{l0} \in \mathcal{S}_{\text{noisy}} \cap \mathcal{Z}^- \end{cases}, \tag{13}$$

where $c = 0$ indicates the observed negative class; $\hat{Y}^{l0}$ represents model prediction for label $l$ and observed class 0.

$$\tilde{Y}_{\text{corr}}^{l1} = \begin{cases} \tilde{Y}^{l1}, & \tilde{Z}^{l1} \in \mathcal{S}_{\text{clean}} \\ Y_{\text{soft}}^{l1}, & \tilde{Z}^{l1} \in \mathcal{S}_{\text{unce}} \\ \hat{Y}^{l1}, & \tilde{Z}^{l1} \in \mathcal{S}_{\text{noisy}} \end{cases}, \tag{14}$$

where $c = 1$ indicates the observed positive class; $\hat{Y}^{l1}$ represents model prediction for label $l$ and observed class 1; $Y_{\text{soft}}^{lc}$ denotes the expected soft label refined by the uncertainty-aware correction strategy (see Equation 12).

## A.5 MODEL IMPLEMENTATION AND HYPERPARAMETER

All experiments are performed on the High-Performance Computing infrastructure using PyTorch 1.11.0 and an NVIDIA A100 GPU. To maintain fairness in comparisons, we apply consistent hyperparameter settings and neural network architecture across all experiments. Early stopping is not used, as we assume the unavailability of a clean validation set, reflecting real-world conditions.

---

**Algorithm 1** MIRACL: Multi-modal Instance Relabelling And Correction for multi-Label noise

---

1: **Input:** Multi-label Dataset $\mathcal{D}$, learning rate $\eta$, max epochs $T$, warmup time $t_{warmup}$
2: **Output:** Trained Model $M$
3: Initialise model $M$
4: **for** epoch $t = 1$ to $T$ **do**
5:     **if** $t \leq t_{warmup}$ **then**
6:         Update $\mathcal{L}_{\text{warmup}}$
7:     **else**
8:         Calibrate $\tilde{Y}$ using Equation (13), (14)
9:         Update $\mathcal{L}_{\text{weighted}}$ with $\tilde{Y}_{\text{corr}}$ by Equation (11)
10:     **end if**
11:     **if** $t \geq t_{warmup}$ **then**
12:         **for** $l = 1$ to $C$ **do**
13:             **for** $c = 0$ to $1$ **do**
14:                 Fit GMM$^{lc}$, select $\mathcal{S}_{\text{clean}}, \mathcal{S}_{\text{unce}}, \mathcal{S}_{\text{noisy}}$
15:             **end for**
16:         **end for**
17:     **end if**
18: **end for**
19: **return** Trained Model $M$

---

### A.5.1 IMPLEMENTATION DETAIL

Each baseline model is trained independently with the same hyperparameter settings. Each model is trained for 30 epochs using the Adam optimizer, with an initial learning rate of $10^{-3}$ scheduled via cosine annealing ($T_{\text{max}} = 10$, $\eta_{\text{min}} = 0$), batch size of 128, a warm-up period of 5, a correlation threshold of $\tau = 0.02$, a regularization strength coefficient $\lambda_{\text{cons}} = 0.1$, and a selection metric coefficient of $\alpha = 0.5$. For each noise type, experiments are repeated three times with three different random seeds $= \{30, 40, 100\}$. To prevent overfitting to corrected labels, we apply a weight decay parameter of $1e^{-5}$ when initiating label correction. The noise ratios are defined as follows: $\rho_-, \rho_+ \in \{0.2, 0.4\}$ for Symmetric; Asymmetric Flip Noise $\rho_-, \rho_+ \in \{0, 0.2, 0.4\}$; Balanced Noise $\rho_+ = \rho \in \{0.2, 0.4\}, \rho_- = \{0.0448, 0.0896\}$ respectively for MIMIC-IV phenotyping, $\rho_- = \{0.0395, 0.0790\}$ for MIMIC-III phenotyping and $\rho_- = \{0.0382, 0.0764\}$ for diagnosis.

### A.5.2 BASELINE SETTING

We use FlexCare Xu et al. (2024a) as the backbone model for noisy multi-label approaches and compare our approach to several baseline models with same hyperparameter setting:

- **FlexCare** Xu et al. (2024a): layers=4, expert_k=2, expert_total=10, hidden_dim = 128, ehr_dim = 76, max-length = 512

- **Focal Loss (Focal)** Lin et al. (2017): Focusing parameter $\gamma = 2.0$, Alpha-balancing weight $\alpha = 0.25$

- **Asymmetric Focal Loss (ASL)** Ridnik et al. (2021b): Negative focusing parameter $\gamma_- = 4.0$, Positive focusing parameter $\gamma_+ = 1.0$, Probability margin (clipping) $m = 0.05$,

- **Generalised Cross-Entropy (GCE)** Zhang & Sabuncu (2018): Default parameters, designed to be robust to noise.

- **MLLSC** Ghiassi et al. (2023): Positive threshold $\tau_{pos} = 0.55$, Negative threshold $\tau_{neg} = 0.6$, Margin $m = 1.0$, Gamma $\gamma = 2.0$

- **MultiT** Li et al. (2022b): Default parameters, designed to perform loss correction based on estimated transition matrix $\hat{T}$.

- **M3Care** Zhang et al. (2022): hidden_dim = 128, ehr_dim = 76, dropout = 0.1

- **MedFuse** Hayat et al. (2022): hidden_dim = 128, ehr_dim = 76, dropout = 0.1

Table 6: Computation time comparison for different models under Sym. 20% condition. Computation time (h) refers to the time for a single run

| Model Name | Computation Time (h) |
|---|---|
| MedFuse | 3.108 |
| MultiT | 3.979 |
| M3Care | 3.118 |
| FlexCare | 3.884 |
| MIRACL (Ours) | 4.104 |

### A.6 COMPUTATIONAL ANALYSIS

Despite incorporating $L \times 2$ Gaussian Mixture Model (GMM) for dynamic sample selection, MIRACL does not introduce significant computational since it does not rely on re-training strategy against corrected labels. As shown in Table 6, its total training time remains comparable to other advanced baselines. This efficiency stems from our design choice to fit the GMM once ***per epoch rather than per batch***, and only after the warm-up phase, which amortizes the cost and avoids redundant computation. This demonstrates that MIRACL achieves robust noise correction with gradual increase in training time. **Notably, most of the time complexity stems from the underlying Flex-Care backbone shared by MIRACL, rather than the noise modeling module itself.**

### A.7 OTHER RELATED WORKS

Label noise under EHR is gaining more attention recently. Initial efforts in addressing single-label noise underlying a neighbor consistency regularization approach in unimodal EHR Yang et al. (2024). MEDFuse Phan et al. (2024) presents a LLM-enhanced multimodal EHR fusion framework with masked lab-test modeling. While their method emphasizes imputation and masked recovery with LLMs, our framework instead targets label denoising with a model-agnostic backbone, offering complementary contributions. Zhan et al. (2023) introduces a reliability-based cleaning method using inductive conformal prediction, which shares our goal of identifying trustworthy samples in noisy multimodal contexts. However, MIRACL further integrates label ranking and modality-specific difficulty into the sample selection pipeline. Li et al. (2025) dynamically augments and calibrates labels in EHRs by modeling temporal uncertainty under time series data. In contrast, MIRACL performs static and progressive correction via joint relabeling, and explicitly accounts for cross-modal inconsistencies rather than purely temporal calibration. In the image domain, BoMD Chen et al. (2023) introduces descriptor-based re-labeling for noisy multi-label classification in chest X-rays. While BoMD focuses on image-noise structure, our work addresses multimodal fusion with textual and temporal signals and supplements with patient-level contrastive regularization during correction.

Contrastive learning offers a compelling solution by enforcing alignment through multimodal instance or class-level objectives. While supervised contrastive learning (SCL)Chen et al. (2020) has shown strong performance, recent efforts extend it into noisy Li et al. (2022a) and medical domains Wang et al. (2022). However, there is still a critical gap: no prior work has adapted patient-level contrastive learning for the unique pairing of structured EHRs and clinical notes, nor has it been investigated as a mechanism to mitigate label noise by leveraging longitudinal patient context.

## B ADDITIONAL ANALYSIS AND VISUALIZATIONS

### B.1 SELECTION METRIC ANALYSIS

Figure 6 presents a comparative analysis of BCE loss, ranking, and overall difficulty across training epochs for clean and noisy instance-label pairs under symmetric $(40\%, 40\%)$ label noise. Among the three, memorization-based metrics (Fig. 1c) demonstrate the strongest discriminative power during the early training phase (e.g., epochs 0–40), where the curves for clean and noisy pairs—both positive and negative—are clearly separable. This behavior aligns with the prior observation that deep networks tend to fit clean samples earlier.

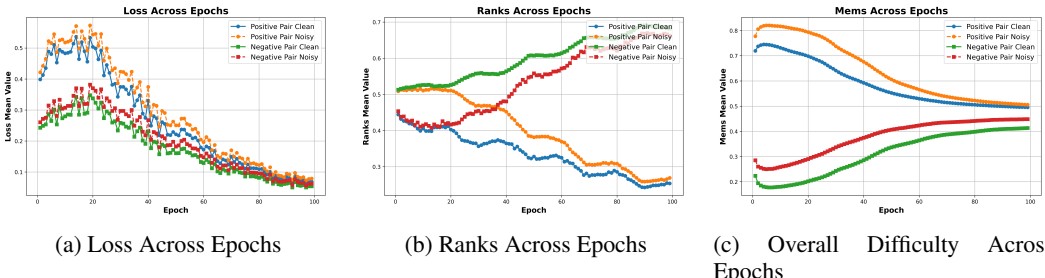

(a) Loss Across Epochs     (b) Ranks Across Epochs     (c) Overall Difficulty Across Epochs

Figure 6: Comparison of metrics such as BCE Loss, Ranks, and Overall Difficulty across 100 training epochs using the Vanilla FlexCare model under 5% of MIMIC-Phenotyping Dataset with Symmetric (20%, 20%) flip noise.

Table 7: Statistical comparison of MIRACL vs. second-best baseline under each noise setting (based on test mAP over 3 runs) on MIMIC-IV phenotyping.

| Noise Type $(\rho_+, \rho_-)$ | Second-best | $p$-value | Significance |
|---|---|---|---|
| (20,20) | GCE | 0.00082 | *** |
| (40,40) | GCE | 0.00288 | ** |
| (0,20) | MultiT | 0.00807 | ** |
| (0,40) | FlexCare | 0.01233 | * |
| (20,0) | MultiT | 0.05908 | |
| (40,0) | FlexCare | 0.03163 | * |
| (20,4.48) | FlexCare | 0.00002 | *** |
| (40,8.96) | MultiT | 0.00367 | ** |

In contrast, ranking-based indicators (Fig. 1b) become more stable and reliable in later training stages (after epoch 50), consistently assigning higher ranks to clean positive labels while maintaining a steady gap between clean and noisy pairs. This suggests that rank-based selection becomes more robust once the model has formed high-confidence predictions.

These observations validate our two-stage design: leveraging difficulty metric in the early phase to identify clean pairs based on learning dynamics, and adopting rank-based metrics in the later phase to exploit the model's matured confidence estimates.

## B.2 SENSITIVITY ANALYSIS FOR GMM INITALIZATION

GMM does have some dependence on initialization, but in our setting the effect is very small. We evaluated the symmetric-20initializations and observed only minor fluctuations in performance. The coefficient of variation (CV) across runs is below 1(mAP: 0.61%, F1: 0.73%, F1_class: 0.98%), and the score ranges are narrow (e.g., mAP varies from 0.5433 to 0.5499). These results show that the selection scores form a stable bimodal structure, so different GMM initializations lead to nearly identical clean/noisy assignments. The small run-to-run variance suggests that GMM initialization does not materially affect MIRACL's robustness.

## B.3 BIMODAL ASSUMPTION VERIFICATION

Fig. 7 illustrates the selection-metric distributions and GMM fits for the five rarest phenotypes in MIMIC-IV under Sym. 20% setting. These conditions represent the most challenging scenarios for identifying clean versus noisy samples due to extremely low prevalence and high label imbalance. For each phenotype—Acute cerebrovascular disease, Acute myocardial infarction, Gastrointestinal hemorrhage, Other upper respiratory disease, and Pleurisy/Pneumothorax/Pulmonary collapse—the empirical density displays a distinct two-mode structure. The dominant mode corresponds to easy-to-learn (clean) samples with low selection scores, while a secondary, smaller mode captures harder or potentially noisy samples.

Table 8: Statistical comparison of MIRACL vs. second-best baseline under each noise setting (based on test mAP over 3 runs) on MIMIC-III phenotyping.

| Noise Type $(\rho_+, \rho_-)$ | Second-best | $p$-value | Significance |
|---|---|---|---|
| (20,20) | ASL | 0.00610 | ** |
| (40,40) | ASL | 0.98795 | |
| (0,20) | ASL | 0.00057 | *** |
| (0,40) | ASL | 0.00068 | *** |
| (20,0) | MultiT | 0.00328 | ** |
| (40,0) | FlexCare | 0.01053 | * |
| (20,3.95) | MultiT | 0.00771 | ** |
| (40,7.90) | MultiT | 0.01376 | * |

| Metric | Mean $\pm$ Std | CV | Range |
|---|---|---|---|
| mAP | $0.5468 \pm 0.0033$ | 0.61% | 0.5433–0.5499 |
| F1 | $0.5794 \pm 0.0042$ | 0.73% | 0.5755–0.5839 |
| F1_Class | $0.4349 \pm 0.0043$ | 0.98% | 0.4305–0.4390 |

Table 9: Stability of MIRACL under different GMM initialization seeds. The table reports the mean, standard deviation, coefficient of variation (CV), and value range across five runs. The low standard deviations and CV ($< 1\%$ for all metrics) indicate that MIRACL's GMM-based selection is highly robust to initialization.

Across all five cases, the fitted 2-component GMMs clearly separate these two regimes with strong component separation, consistent BIC improvements, and silhouette scores around 0.68. These results provide empirical evidence that the bimodal assumption underlying MIRACL's class-aware correction remains valid even for rare phenotypes with severe class imbalance.

## C  FULL QUANTITATIVE RESULTS & CHECKLIST

### C.1  MIMIC-IV DIAGNOSIS EXPERIMENT:

While MIRACL demonstrates state-of-the-art performance on the phenotyping task, our results from Table 10 show that it does not consistently outperform M3Care in the diagnosis setting. This discrepancy is attributable to the distinct architectural priorities of the two models in the face of extreme modality missingness. The diagnosis dataset suffers from severe data sparsity, with 76.3% of time-series and 32.6% of clinical notes absent. M3Care is explicitly designed to handle this challenge through robust modality-specific pathways and dropout mechanisms. In contrast, MIRACL's core strength lies in leveraging cross-modal signals for label noise correction. When one or both modalities are frequently absent, MIRACL's ability to cross-reference evidence is fundamentally limited, reducing its advantage. Nevertheless, MIRACL consistently ranks as the second-best model across most noise configurations, indicating strong generalization despite missing data. This analysis underscores that robustness to label noise and robustness to missing modalities are distinct challenges, and MIRACL is highly specialized for the former. Future work could explore hybrid architectures that combine MIRACL's sophisticated label correction with M3Care's proven robustness to missing data.

### C.2  STATISTICAL TESTING

We perform one-sided Student's $t$-tests (across 3 runs) comparing MIRACL to the second-best baseline under each noise condition on MIMIC-III (Table 8) Phenotyping and MIMIC-IV Phenotyping (Table 7). Significance is marked in the table using *, **, and ***, indicating $p<0.05$, $p<0.01$, and $p<0.001$, respectively. All tests compare test mAP scores under the same seeds.

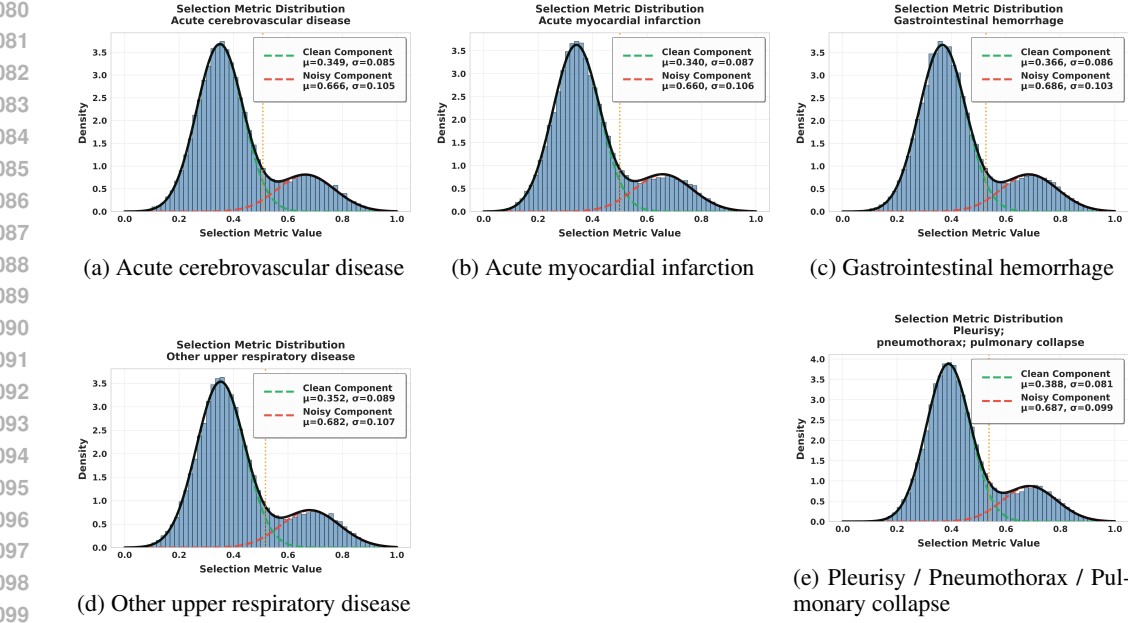

(a) Acute cerebrovascular disease    (b) Acute myocardial infarction    (c) Gastrointestinal hemorrhage

(d) Other upper respiratory disease

(e) Pleurisy / Pneumothorax / Pulmonary collapse

Figure 7: Selection metric distributions and 2-component GMM fits for the five rarest phenotypes in MIMIC-IV under Symm. 20% setting. All cases show a clear bimodal structure separating easy-to-learn (clean) and hard-to-learn (noisy) regimes, supporting the GMM assumption used in MIRACL's class-aware correction module.

Table 10: Comparison of performance on MIMIC-IV Diagnosis test dataset under different noise conditions ($\rho_+$, $\rho_-$). The evaluation metric is average mAP with standard deviation (in bracket) in the last epoch across 3 runs.

| Model | Symmetric Flip Noise (%) | | Asymmetric Flip Noise (%) | | | | Balanced Noise (%) | |
|---|---|---|---|---|---|---|---|---|
| | (20,20) | (40,40) | (0,20) | (0,40) | (20,0) | (40,0) | (20,3.82) | (40,7.64) |
| ASL | 0.207(0.006) | 0.197(0.004) | 0.221(0.003) | 0.211(0.004) | 0.198(0.010) | 0.197(0.013) | 0.202(0.010) | 0.196(0.005) |
| Focal | 0.18(0.000) | 0.167(0.011) | 0.18(0.000) | 0.174(0.011) | 0.173(0.012) | 0.167(0.011) | 0.167(0.011) | 0.18(0.000) |
| GCE | 0.193(0.006) | 0.193(0.003) | 0.208(0.004) | 0.195(0.007) | 0.193(0.004) | 0.189(0.003) | 0.19(0.001) | 0.193(0.002) |
| MLLSC | 0.157(0.007) | 0.154(0.002) | 0.153(0.002) | 0.157(0.006) | 0.159(0.006) | 0.159(0.006) | 0.157(0.006) | 0.159(0.006) |
| MultiT | 0.2(0.004) | 0.193(0.003) | 0.218(0.006) | 0.195(0.002) | 0.23(0.001) | 0.22(0.016) | 0.214(0.021) | 0.196(0.009) |
| M3Care | **0.219(0.000)** | 0.206(0.000) | 0.222(0.000) | **0.22(0.000)** | 0.224(0.000) | 0.223(0.000) | 0.224(0.000) | **0.22(0.000)** |
| MedFuse | 0.208(0.001) | 0.195(0.001) | 0.214(0.002) | 0.212(0.001) | 0.218(0.004) | 0.217(0.001) | 0.216(0.003) | 0.209(0.001) |
| FlexCare | 0.194(0.007) | 0.194(0.004) | 0.214(0.009) | 0.212(0.010) | **0.231(0.001)** | 0.219(0.017) | 0.209(0.013) | 0.198(0.008) |
| MIRACL | **0.219(0.000)** | **0.207(0.002)** | **0.223(0.001)** | **0.22(0.001)** | 0.228(0.001) | **0.225(0.001)** | **0.225(0.002)** | 0.219(0.001) |

## D    TECHNICAL DETAILS OF LABEL CORRELATION MATRIX

We construct the label correlation matrix $\mathbf{C} \in \mathbb{R}^{L \times L}$ using label co-occurrence statistics across the training set. Each entry $C_{k,j}$ reflects the normalized co-occurrence frequency between label $k$ and label $j$, defined as:

$$C_{k,j} = \begin{cases} 0, & k = j \\ \frac{\sum_{i=1}^{N} \tilde{Y}_{i,k} \cdot \tilde{Y}_{i,j}}{\sum_{b=1}^{L} \sum_{i=1}^{N} \tilde{Y}_{i,k} \cdot \tilde{Y}_{i,b}}, & k \neq j \end{cases}$$

where $\tilde{Y}_{i,j} \in \{0, 1\}$ indicates whether the $j$-th label is assigned to the $i$-th instance. The row normalization ensures that each row of $\mathbf{C}$ sums to 1, facilitating a probabilistic interpretation of inter-label dependency.

