# OpenReview forum: "MIRACL: A Robust Framework for Multi-Label Learning on Noisy Multimodal Electronic Health Records"
_ICLR.cc/2026/Conference — ICLR 2026 Conference Withdrawn Submission_

### Official Review · Reviewer_3B3W · 2025-10-21

**Soundness:** 2
**Presentation:** 1
**Contribution:** 2
**Rating:** 4
**Confidence:** 3

**Summary:**

This paper presents MIRACL, a framework designed to address label noise in Electronic Health Records (EHRs) by combining difficulty-based noise detection, label correction, and patient-level contrastive learning. The model incorporates both structured (e.g., time-series medical records) and unstructured (e.g., clinical notes) data, with a focus on multi-label learning. Experiments on the MIMIC-III and MIMIC-IV datasets show that MIRACL outperforms state-of-the-art methods.

**Strengths:**

- The framework combines difficulty-based noise detection, label correction, and patient-level contrastive learning, which makes it a strong contender for noisy label problems in healthcare.
- By handling both structured and unstructured data, MIRACL significantly enhances the utility of EHRs, which are inherently multimodal in nature.

**Weaknesses:**

- The simplified label noise types (e.g., random label flipping) used in the experiments may not capture the dependencies or temporal patterns of noise typically observed in healthcare data. I believe the authors should redesign the way noise is introduced into the data.

- There is a lack of related baselines, such as ACTLL [1], STW [2], SREA [3], and SIGUA [4], for comparison with MIRACL on the MIMIC-III and MIMIC-IV datasets.

- The paper lacks more benchmark datasets, such as the eICU Collaborative Research Database [5].

- The figures in the paper need to be polished. There are some issues with the lines and arrows.

[1] Dynamical Label Augmentation and Calibration for Noisy Electronic Health Records.

[2] CTW: Confident Time-Warping for Time-Series Label-Noise Learning.

[3] Estimating the electrical power output of industrial devices with end-to-end time-series classification in the presence of label noise.

[4] Sigua: Forgetting may make learning with noisy labels more robust.

[5] The eICU Collaborative Research Database, a freely available multi-center database for critical care research

**Questions:**

See weaknesses.

---

> ### Author Response · Authors · 2025-11-28
>
> > **W1: The simplified label noise types (e.g., random label flipping) used in the experiments may not capture the dependencies or temporal patterns of noise typically observed in healthcare data. I believe the authors should redesign the way noise is introduced into the data.**
> >
>
> We appreciate the reviewer’s concern regarding simplified noise models. Our chosen noise types (symmetric, asymmetric, and balanced label flips) are **standard and widely adopted** in robust multi-label learning literatures because they offer a **controlled, reproducible, and comparable** benchmark for evaluating robustness across methods. Although simplified, these settings span diverse and challenging corruption patterns that sufficiently **stress-test MIRACL’s resilience**.
>
> We agree that clinically induced or temporally dependent noise is an important direction. But its beyond the scope of this paper. We will clarify this and highlight temporal label noise as meaningful **future work**.
>
> > **W2  There is a lack of related baselines, such as ACTLL, STW, SREA, and SIGUA, for comparison with MIRACL on the MIMIC-III and MIMIC-IV datasets.**
> >
>
> We thank the reviewer for suggesting additional baselines. However, the proposed methods(ACTLL[1], CTW[2], SREA[3], SIGUA[4]) are **single-label, unimodal time-series noisy-label approaches**, whereas phenotyping in MIMIC-III/IV is a **multimodal (EHR + notes), multi-label** task with **4.1–4.6 positive labels per visit**. These methods assume:
>
> - one ground-truth label per instance
> - unimodal sequential input
>
> These assumptions do not hold in our setting, and the methods cannot be used directly without substantial redesign.
>
> > **W3: The paper lacks more benchmark datasets, such as the eICU Collaborative Research Database [5].**
> >
>
> We appreciate the suggestion. However, the eICU Collaborative Research Database[5] currently **does not provide a well-defined multi-label learning task**, and directly adapting MIRACL without a standardized label set would limit comparability.
>
> > **W4 The figures in the paper need to be polished. There are some issues with the lines and arrows.**
> >
>
> We thank the reviewer for pointing this out. **Figures 1 and 2 have been fully polished** in the revised version, including corrected line styles, arrow placements, and overall visual clarity.

---

### Official Review · Reviewer_mCUm · 2025-10-24

**Soundness:** 3
**Presentation:** 1
**Contribution:** 2
**Rating:** 4
**Confidence:** 5

**Summary:**

The paper proposes MIRACL, a framework designed to address multi-label noise in multimodal Electronic Health Records (EHRs). It proposed to utilize three modules to mitigate label noise: a multi-faceted selection metric, a class-aware correction module, and a patient-level contrastive regularization loss. Experimental results on MIMIC-III/IV datasets show that MIRACL achieves state-of-the-art robustness under various noise settings, demonstrating its effectiveness, particularly in high-noise regimes.

**Strengths:**

1. **Important and Practical Topic**: The multi-label noise in healthcare is important, and this topic has the potential to make an impact in the real world.
2. **Comprehensive Framework for Multi-Label Noise**: MIRACL is presented as the comprehensive framework unifies three synergistic mechanisms to achieve robustness: class-wise sample selection, label correction, robust contrastive learning.
3. **State-of-the-Art Performance in Phenotyping**: Extensive experiments on MIMIC-III and MIMIC-IV Phenotyping (PHE) datasets demonstrate that MIRACL achieves state-of-the-art robustness

**Weaknesses:**

1. **Limited Novelty of The Proposed Solution**: It is more like a mixture of many existing methods for sample selection, label correction, and contrastive learning.
2.  **Limited Effectiveness of Selection Metric**: As shown in Table 4, the Correction w/ Loss only, w/ Mems only, w/ Rank only do not perform significantly worse than MIRACL.
3. **Not Good Presentation**:
(1) For example, visits and patients are not defined in problem formulation. In Line 193, $\Delta^{(t)}_f$ is not defined.
(2) In Line 204, why can instance-level prediction dynamics be used to capturing inter-label dependencies?
(3) In Line 215, what is 2 x L?
(4) In Line 255,  $\mathbb{E}$ usually refers to the expectation and can be easily misunderstanded here.
(5) In Line 268, how to compared the correlation matrix with a threshold?

**Questions:**

In Line 433, the statement that the performance curve shows a clear upward trend as $λ_{cons}$ increases from 0.0 to 0.05 lacks evidence.

---

> ### Author Response · Authors · 2025-11-28
>
> > **W1: Limited Novelty of The Proposed Solution: It is more like a mixture of many existing methods for sample selection, label correction, and contrastive learning.**
> >
>
> We thank the reviewer for raising this point. MIRACL is not a simple combination of existing components; instead, it introduces **new synergies that do not appear in prior work**:
> As shown in **Table 4**, the full MIRACL model consistently outperforms all single-component variants. None of the individual modules (loss-only, memorization-only, rank-only) can match the robustness of the integrated design. This pattern indicates that the improvement arises from the **interaction** among components rather than from any single mechanism, demonstrating that MIRACL provides more than a simple mixture.
>
> > **W2: Limited Effectiveness of Selection Metric: As shown in Table 4, the Correction w/ Loss only, w/ Mems only, w/ Rank only do not perform significantly worse than MIRACL.**
> >
>
> Each individual metric (loss, memorization, rank) indeed provides a meaningful signal, which naturally limits the standalone performance gap. MIRACL, however, is designed around **complementarity**, not reliance on any single cue. The metrics capture **orthogonal aspects** of instance–label difficulty—memorization reflects learning and forgetting dynamics across epochs, while rank measures relative confidence ordering within each visit.
>
> As shown in **Figure 5**, the combined selection metric exhibits a much clearer bimodal structure than any component alone. The clean and noisy labels fall into more distinctly separated regions, which yields more stable thresholds and more reliable correction. In contrast, single-signal variants show greater overlap between modes, particularly under high-noise and multimodal conditions. This explains why the integrated metric leads to more consistent clean/noisy separation and ultimately stronger robustness.
>
> > **W3.1: Presentation For example, visits and patients are not defined in problem formulation.**
> >
> Please check updated problem formulation.
> > **W3.2: Presentation** $\Delta_f^{(t)}$ **is not defined**
> >
> We define **$\Delta_f^{(t)}(\hat{y}_{i}^l)$** = 1 if  $\hat{y}_{i}^{\,l}$ is correct at epoch $t-1$ but incorrect at epoch $t$,
> indicating a **forgetting event**.
>
> > **W3.3: Presentation In Line 204, why can instance-level prediction dynamics be used to capturing inter-label dependencies?**
> >
>
> Instance-level prediction dynamics reflect how the model learns and forgets *each* label for the *same visit* across epochs. In multi-label phenotyping, labels that frequently co-occur (e.g., CHF ↔ Atrial fibrillation, Sepsis ↔ Pneumonia) create correlated gradients during training. When the model updates one label, the shared representation affects its correlated labels as well.
>
> As a result, the **learning/forgetting patterns of one label are not independent**—they are influenced by the presence or absence of other clinically related labels.
>
> By tracking these dynamics over epochs, MIRACL’s metric indirectly captures:
>
> - shared representation effects
> - co-occurrence structure of phenotypes
> - when labels behave inconsistently with their correlated labels (often a sign of noise)
>
> This is why prediction dynamics at the instance level contain meaningful signal about **inter-label dependencies**, even though we do not model a label–label graph explicitly.
>
> > **W3.4: Presentation In Line 215, what is 2 x L?**
> >
>
> We explain that **“2 × L”** refers to fitting one **2-component GMM for each label** and each **class value**   $c\in \{0,1\}$.
>
> > **W3.5: Presentation In Line 255,
> usually refers to the expectation and can be easily misunderstanded here.**
> >
>
> We agree that the notation $\mathbb{E}[\cdot]$ may be misinterpreted as an expectation operator.
> We have replaced it with a neutral notation for corrected soft labels:
>
> $Y_{\text{soft}}$ as shown in updated Equation 5
>
> > **W3.6: Presentation In Line 268, how to compared the correlation matrix with a threshold?**
> >
>
> We thank the reviewer for pointing this out. Following by iLaCO[1], we clarify that the thresholding is
> not applied to the entire correlation matrix $C$. The dimensions involved are:
>
> - $Y \in \mathbb{R}^{N \times L}$: the (noisy) multi-label matrix for all $N$ instances
> - $C \in \mathbb{R}^{L \times L}$: the label–label correlation matrix
> For each instance $i$, we compute its correlation-augmented label score:
>
> $S = \tilde{Y} \ C$
>
> where $S$ reflects how strongly each label is supported by correlated labels in the prediction space. This has been shown in Equation 9,
>
> This operation propagates support from correlated labels and yields a refined score
> for each of the $L$ labels.
>
>
> > **Questions**
> >
>
> We appreciate the reviewer’s question. We clarify that “the performance curve shows a clear upward trend ***under high noise scenarios*** as increases from 0.0 to 0.05.”
>
> [1] Noisy Multi-Label Text Classification via Instance-Label Pair Correction

---

### Official Review · Reviewer_gAtv · 2025-11-01

**Soundness:** 3
**Presentation:** 2
**Contribution:** 2
**Rating:** 2
**Confidence:** 4

**Summary:**

This paper tackles multi-label noise in multimodal EHR data combining structured time-series and unstructured notes. The proposed framework, MIRACL, unifies three components: (1) difficulty- and rank-based detection of noisy labels, (2) class-aware label correction, and (3) patient-level contrastive regularization leveraging longitudinal context. Experiments on MIMIC-III/IV show consistent gains under various noise settings.

**Strengths:**

- The paper addresses an important and realistic problem of noisy labels in multimodal, longitudinal EHRs.

- The framework is well-structured and empirically validated under multiple corruption types and noise levels.

- The clinical multimodal scenario is meaningful and underexplored.

**Weaknesses:**

- The claim of being “the first to leverage longitudinal context” appears overstated given prior work on temporal or contextual consistency.

- The added difficulty of multimodality vs. single-modality noise learning is not clearly explained.

- The literature review and motivation are weak; for example, why existing methods (e.g., BalanceMix) are not applicable is not convincingly argued.

- The three modules borrow heavily from prior methods, lacking clear methodological novelty.

- The baseline comparison is insufficient, omitting recent multimodal fusion and noisy-label learning approaches.

**Questions:**

1. What unique challenges does multimodal label noise introduce beyond the single-modality case?

2. How is longitudinal context quantitatively used in the contrastive module?

3. Minor typo: Figure 2: “Atrraction” -> “Attraction.”

---

> ### Author Response · Authors · 2025-11-28
>
> > **Weakness 1: The claim of being “the first to leverage longitudinal context” appears overstated given prior work on temporal or contextual consistency.**
> >
>
> We thank the reviewer and agree the wording can be improved.
>
> Our intention was not to claim novelty in *temporal consistency* in general, but to highlight that **no prior noisy-label method for multimodal EHRs uses patient-level longitudinal evidence for noise resolution**, which is the core of our contribution.
>
> > **Weakness 2: The added difficulty of multimodality vs. single-modality noise learning is not clearly explained.**
> >
>
> Multimodal label noise is more difficult
> than single-modality noise because noise interacts with the *modality gap* [1]—the
> mismatch between heterogeneous clinical signals such as structured time-series
> features and unstructured notes. After fusion, these modalities form a joint
> representation. When the label is incorrect, the two modalities often support
> different clinical interpretations, and the fused embedding is forced to
> integrate inconsistent evidence. This amplifies the impact of noise because it
> disturbs not only one modality but also the alignment between modalities, a
> challenge that does not occur in single-modality settings.
>
> > **Weakness 3: The literature review and motivation are weak; for example, why existing methods (e.g., BalanceMix) are not applicable is not convincingly argued.**
> >
>
> Our Appendix A.7 has already included an extended literature discussion.
>
> ### **Why BalanceMix is not directly applicable**
>
> As our Related Work section notes (Sec. 2)  :
>
> - BalanceMix is designed for **image-space Mixup augmentation**, relying on pixel-level convex combinations and label smoothing that do **not transfer** to heterogeneous EHR modalities (structured time-series + text).
> - No official implementation exists, and the method assumes continuous visual spaces.
>
>     Multimodal EHR inputs (irregular multivariate time-series + token embeddings) do not satisfy these assumptions.
>
>
> > **Weakness 4: The three modules borrow heavily from prior methods, lacking clear methodological novelty.**
> >
> We thank the reviewer for raising this point. MIRACL is not a simple combination of existing components; instead, it introduces new synergies that do not appear in prior work: As shown in Table 4, the full MIRACL model consistently outperforms all single-component variants. None of the individual modules (loss-only, memorization-only, rank-only) can match the robustness of the integrated design. This pattern indicates that the improvement arises from the interaction among components rather than from any single mechanism, demonstrating that MIRACL provides more than a simple mixture.
>
> > **Weakness 5: The baseline comparison is insufficient, omitting recent multimodal fusion and noisy-label learning approaches.**
> >
>
> Our current baselines include: **Multimodal EHR fusion models, State-of-the-art noisy multi-label methods.**
>
> We selected baselines that are:
>
> - reproducible, and
> - compatible with multimodal EHR pipelines
>
> As discussed in Sec. 2, many recent multimodal fusion or noisy-label methods lack publicly available implementations or are incompatible with heterogeneous EHR modalities.
>
> > **Question 1: What unique challenges does multimodal label noise introduce beyond the single-modality case?**
> >
>
> Please see answer for Weakness 2
>
> > **Question 2: How is longitudinal context quantitatively used in the contrastive module?**
> >
>
> MIRACL leverages longitudinal structure by using multiple visits from the same
> patient as a consistency signal to regularize multimodal embeddings. While the
> method does not model temporal ordering or sequential trajectories, it exploitsL
> the repeated-visit structure of EHRs to capture stable patient-level patterns,
> which significantly improves robustness to noisy labels.
>
> For example, if a patient has visits at $t=1$ and $t=2$ that share similar
> structured features (e.g., consistent creatinine levels) and similar text
> descriptors (e.g., “history of hypertension”), the contrastive objective pulls
> their embeddings closer even if one of the labels is noisy or partially missing.
> Conversely, if a later visit introduces new evidence (e.g., “new onset heart
> failure” appearing in notes), the encoder still produces a distinct embedding
> because the underlying multimodal features differ. Thus, longitudinal context is
> used quantitatively through patient identity.
>
> > **Question 3:**  Minor typo: Figure 2: “Atrraction” -> “Attraction.”
> >
>
> Thank you for the suggestion. We have already fixed this typo in our revised version.
>
> [1] Mind the Gap: Understanding the Modality Gap in Multi-modal Contrastive Representation Learning

---

### Official Review · Reviewer_3LPu · 2025-11-04

**Soundness:** 2
**Presentation:** 2
**Contribution:** 2
**Rating:** 2
**Confidence:** 4

**Summary:**

This paper introduces MIRACL, a multimodal EHR framework for handling multi-label noise, integrating three complementary components: a sample selection strategy based on difficulty and ranking metrics combined with GMM; a label correction mechanism designed for different sample categories (clean/uncertain/noisy); and patient-level contrastive regularization utilizing patient longitudinal context. This framework addresses multiple noise issues and achieves consistent performance improvements on the MIMIC-III/IV datasets.

**Strengths:**

I find the most interesting merit of this paper lies in its design for forgetting and remembering multi-labels, particularly how MIRACL tracks the learning and forgetting behavior of each multi-label during every training epoch. Additionally, the ranking design aids in understanding label credibility.

**Weaknesses:**

The overall writing structure of the paper lacks clarity. I have several comments:

1. I find that the paper's claimed longitudinal design has fundamental issues. For example, it pulls all visits of the same patient closer together but does not model temporal ordering. True longitudinal modeling should involve sequential trajectories: $\text{Visit}_1 \rightarrow \text{Visit}_2 \rightarrow \text{Visit}_3$.

2. There is a logical inconsistency in the paper. The premise states that MIMIC data may contain noise (motivation), yet the experimental setup assumes MIMIC is clean, and the conclusion claims that the method can handle noise. This reasoning is not self-consistent.

3. Consider the following situation. For instance, take Patient A: Visit~1 in~2020 shows Diabetes= 0 (healthy), whereas Visit~2 in~2023 shows Diabetes=1 (newly diagnosed). However, MIRACL's approach of forcing $h_{\text{visit1}} \approx h_{\text{visit2}}$ seems unreasonable.

4. Although the memorization and forgetting design in Section 3.3.1 is interesting, how do you ensure this design is effective or accurate for multi-label classification across patients with varying numbers of visits? Especially for patients with only 1 or 2 visits.

5. The positive sample definition is too coarse. It does not distinguish between visit time intervals---1 day later vs. 3 years later. It does not consider disease state changes---for example, should visits for acute conditions (such as pneumonia) be connected? It does not consider label consistency---what if two visits have completely different labels?

6. The synergy with label correction is unclear. How do contrastive learning and label correction mutually reinforce each other? The paper does not elaborate on this in depth.

7. The notation is inconsistent, with $Y$, $\tilde{Y}$, $\hat{Y}$, and $Y_{\text{corr}}$ used interchangeably.

8. According to the paper's experimental setup, are all different visits of the same patient in the training set? If so, new visits in the test set could leverage historical information from training, which appears to be data leakage. What about temporal splitting? The data should be split temporally (e.g., training on pre-2020 data and testing on post-2021 data).

**Questions:**

1. Is GMM fitting sensitive to initialization?
2. Does the bimodal assumption still hold for extremely imbalanced datasets (e.g., certain rare disease labels)?
3. How does patient-level contrastive learning operate when handling new patients with only a single visit record?

---

> ### Author Response · Authors · 2025-11-28
>
> > **W1: longitudinal design**
> >
>
> We appreciate the reviewer’s observation. The purpose of the longitudinal contrastive
> module is different: it leverages within-patient consistency as an additional
> regularization signal to stabilize multimodal representations under label noise.
>
> We agree that integrating explicit temporal structure is an interesting
> direction and could improve MIRACL, but it is not the goal of the current
> framework.
>
> > **W2: Logical Inconsistency**
> >
>
>  Our motivation acknowledges that MIMIC labels may contain real-world noise (e.g., under-coding or documentation delay), but the exact amount and pattern of this noise is unknown and cannot be measured directly.
> This protocol (introducing synthetic noise to the training set only) allows robustness methods to be compared fairly, because the true real-world noise in MIMIC is unknowable.
> We have clarified this assumption in the revised version (line 361).
>
> > **W3: Consider the following situation. For instance, take Patient A: Visit~~1 in~~2020 shows Diabetes= 0 (healthy), whereas Visit~~2 in~~2023 shows Diabetes=1 (newly diagnosed). However, MIRACL's approach of forcing seems unreasonable.**
> >
>
> We agree that different visits from the same patient may reflect genuine changes
> in clinical state (e.g., a new diabetes diagnosis). MIRACL does not force these
> labels to be the same. Label correction is performed strictly at the visit
> level, so Visit 1 retains its label of 0 and Visit 2 retains its label of 1 if MIRACL doesn't detect any label noise. No
> component in MIRACL overrides or equalizes labels across visits. The longitudinal contrastive module operates only on the representation space.
>
> > **W4: the memorization and forgetting design **
> >
>
> The memorization and forgetting metrics are computed per instance–label pair and
> do not rely on the number of visits a patient has.
>
> We acknowledge that forgetting dynamics are inherently weaker for patients with
> only 1–2 visits, which limits the strength of this signal. However, the
> difficulty, memorization, and rank signals are complementary; our ablation
> studies from Table 4 show that the combined metric remains effective even in such settings.
>
> > **W5: Positive Sample Definition.**
> >
>
> Our positive-pair definition does not encode time intervals or assume temporal
> monotonicity. The contrastive module is not designed
> to model disease trajectories, but to provide a patient-level anchor that reduces
> representation variance caused by multimodal inconsistencies and noisy labels.
> Incorporating assumptions about chronic versus acute conditions or time-varying
> disease states would require additional medical priors and structured diagnosis
> timelines, which are often incomplete in MIMIC. We acknowledge this as a
> potential extension.
>
> > **W6: How do contrastive learning and label correction mutually reinforce each other? The paper does not elaborate on this in depth.**
> >
>
> We appreciate the reviewer’s question. The two modules reinforce each other
> because they operate on complementary sources of signal. The label correction
> module improves the quality of the training targets, which reduces the number of
> incorrect supervision signals passed into the encoder.
> Conversely, the contrastive module reduces representation variance that arises
> from multimodal inconsistencies and visit-level noise.
>
> > **W7:  Notation**
> >
>
> We thank the reviewer for pointing this out. We have standardized notation
> throughout the paper:
> - $Y$: latent clean label
> - $\tilde{Y}$: observed noisy label
> - $\hat{Y}$: model prediction
> - $Y_{corr}$: corrected label
>
> > **W8: Data leakage and temporal splitting**
> >
>
> We appreciate the reviewer’s question. Our data split is patient-wise as mentioned in line 357: all visits belonging to a particular patient are assigned entirely to either train,
> or test. This ensures that the model never sees any portion of a
> test patient’s history during training.
>
> Regarding temporal splitting, phenotyping in MIMIC-III/IV is not a forecasting
> task, and the dataset does not represent a synchronized chronological timeline
> across patients. Prior work on this task (e.g., M3Care, MedFuse, and
> multimodal phenotyping baselines) consistently uses patient-wise splits.
>
> > **Q1: GMM fitting**
> >  Yes, moderate sensitivity. Please check the updated Appendix B.2 for more details
>
> > **Q2: Bimodal Assumption**
> > Yes. Please check the updated Appendix B.3 for more details.
>
> > **Q3: How does patient-level contrastive learning operate when handling new patients with only a single visit record?**
> >
>  We acknowledge that the patient-level
> contrastive objective is most beneficial when a patient has multiple visits,
> since the model can draw strength from aligning representations across those
> visits. For patients with only a single visit, this objective naturally
> degenerates into a simple self-consistency term and therefore contributes much
> less signal. We consider this a limitation of the current design.

---

### Note · Authors · 2026-01-09

I have read and agree with the venue's withdrawal policy on behalf of myself and my co-authors.